# Stepwise Removal Process Analysis Based on Layered Corrosion Oxides

**DOI:** 10.3390/ma15217559

**Published:** 2022-10-27

**Authors:** Yuan Ren, Liming Wang, Mingliang Ma, Wei Cheng, Baoli Li, Yuxin Lou, Jianfeng Li, Xinqiang Ma

**Affiliations:** 1Key Laboratory of High Efficiency and Clean Mechanical Manufacture, Ministry of Education, School of Mechanical Engineering, Shandong University, Jinan 250061, China; 2National Demonstration Center for Experimental Mechanical Engineering Education, Shandong University, Jinan 250061, China; 3Laser Institute, Qilu University of Technology (Shandong Academy of Sciences), 3501 Daxue Road, Jinan 250100, China; 4Shandong Qiangyuan Laser of Sdiit Ltd., Liaocheng 252022, China; 5Liaocheng Institutes of Industrial Technology, Liaocheng 252002, China

**Keywords:** engineering machinery, laser cleaning, central composite experimental test, ablation depth, prediction model

## Abstract

The parts of engineering machinery quickly generate rusty oxides in the working process, which seriously affects their service life and safety. How to remove oxides efficiently without damaging the surface of the matrix is a crucial problem. This paper analyzes the critical laser parameters that affect the distribution of material temperature field, which determines the ablation depth of different oxides, by using the central composite experimental design method and taking the surface-ablation depth of Fe_2_O_3_ and Fe_3_O_4_ before and after laser cleaning as response variables to establish the prediction model of single removal volume with the help of Comsol Multiphysics software. The results show a positive correlation between ablation depth and peak power density and a negative correlation with scanning speed. In this process, the experimental results show that the prediction model is natural and effective. A flow chart of laser stepwise cleaning of layered corroded oxides can provide theoretical guidance for the laser cleaning of engineering machinery.

## 1. Introduction

As an essential material in the industry, steel has massive output and demand every year. Q345 steel is widely used in the industrial field given its good strength, electrical conductivity, and all-around performance, especially in engineering machinery [1,2]. However, the working environment is harsh, and the iron matrix is prone to reacting electrochemically with oxygen and water to form rusty oxides. Rusty oxides seriously affect the performance and safety coefficient of parts. The direct loss caused by corrosion is up to hundreds of millions of USD annually [3]. Traditional dedusting technologies include chemical cleaning, high-pressure water-jet cleaning, shot-blasting cleaning, etc. Due to environmental pollution, the rust-returning phenomenon, and operation difficulty, they do not conform to the development path of green cleaning [4,5,6]. As an emerging cleaning technology, considering its advantages of non-contact cleaning, green cleaning, and good controllability, laser-cleaning technology is gradually replacing traditional cleaning technology and is being applied in the industrial field.

Laser-cleaning technology is based on physical–chemical reactions between laser and pollutions, which can remove the attachment from the material surface through ablation, thermal-stress vibration, and the plasma effect [7,8]. Regarding the effective removal of rusty oxides on the surface of materials, scholars have conducted a certain degree of research in the past few decades. For laser cleaning for rusty oxides, Ristic used an Nd: YAG pulsed laser to clean corrosion on the metal surface. It was found that the laser energy density was the critical parameter that affected the cleaning quality, and an energy density that was too small and high would cause poor cleaning quality [9]. Osticioli analyzed the chemical-reaction process between the dirt components in the laser-cleaning process under different pulse widths and found that compared with a long pulse, the short-pulse laser could remove surface rust without damaging the substrate [10]. Ali. compared the effects of 1064 nm and 532 nm lasers on the surface-corrosion removal of low-carbon steel. The results showed that a 1064 nm wavelength laser was more suitable for cleaning the oxide between the cleaning and damage threshold. In this process, the surface roughness and microhardness of the sample were improved after cleaning [11]. Xu investigated the influence law between cleaning effect and laser power, scanning speed, repetition rate, and pulse duration, and the optimized parameters were obtained. Meanwhile, the heat and thermal-stress equations were used to simulate the cleaning effect with the influence of these parameters, which was in good agreement with the experimental data [12]. Scholars used simulated measures to further explore laser cleaning on different materials. Based on Ansys software, Yue investigated the effects of other temporal pulses on micro-tapered slots covered by an oil film, which laid down a theoretical base for actual cleaning work [13]. Lu successfully established a theoretical model of nanosecond laser peeling off the paint. This model could predict the academic cleaning and damage thresholds based on the mechanism of thermal stress [14]. Marimuthu used a two-dimensional transient numerical simulation to study the material-ablation characteristics and substrate thermal effects in laser cleaning of aerospace alloys. In this process, the mechanism of the excimer laser cleaning was proposed [15].

The working environment of mechanical engineering is harsh, and the surface dirt composition of the parts is more complex. A fundamental problem is how to effectively remove rusty oxide without damaging the surface of the matrix. This paper establishes a layered model of laser cleaning based on the attachment composition of the Q345 steel surface. The central composite test was designed to show the single-pulse-removal volume-prediction model of layered materials with the help of Comsol Multiphysics software, which provides theoretical and technical guidance for practical industrial cleaning.

## 2. Layered Model and Numerical Simulation

### 2.1. Simulation Preparation

The oxide layer (Fe_3_O_4_) and the rust layer (Fe_2_O_3_) on the surface of Q345 carbon steel were cleaned by a laser as the research object of the simulation, and the laser-cleaning light source simulated the pulsed-fiber-laser cleaning equipment (QYCL-FP200, Shandong Qiangyuan Laser of Sdiit Ltd., Liaocheng, China). The pulse frequency is adjustable from 10 to 1000 kHz and the pulse width is adjustable from 50 to 500 ns. The spot diameter was set as 0.5 mm, the peak power density of the laser is proportional to the average power of the laser, and the laser-energy density can be changed by controlling the average power of the laser. Based on scanning-electron-microscope and X-ray-diffraction analysis, the attachment composition on the Q345 matrix was determined to be the layered structure of Fe_2_O_3_ and Fe_3_O_4_, as shown in Figure 1 and Figure 2.

### 2.2. Numerical Simulation

Lasers and their materials have three heat processes: conduction, convection, and radiation [16]. Considering the short pulse duration and high peak power for nanosecond lasers, the heat convection and radiation can be neglected in the layered model; heat conduction is the only formation used in Comsol Multiphysics software 6.0.

Comsol Multiphysics software 6.0 was used to draw a two-dimensional layered model, as shown in Figure 3, and the properties of each layer of material are shown in Table 1. The model is divided by local mesh refinement; the model’s surface area with direct laser action is densely gridded, and the grid far from the laser-action area is of average size. This division method can ensure the accuracy of calculation results and improve the computer’s running speed.

The layered geometric model for rust-oxide cleaning is shown in Figure 3, and several assumptions were ensured in the simulation.

(1)The laser energy had Gaussian distribution on the material surface, and only the conduction of laser energy in the Z direction was considered.(2)The material surface was infinitely large, and the oxide layers and matrix were insulated on both sides.(3)L_1_, L_2_, and L_s_ represent the thickness of Fe_2_O_3_, Fe_3_O_4_, and matrix layers, respectively.(4)The Z-direction complied with the absorption law, and each parameter did not change with temperature.

Equations (1)–(6) describe one-dimensional heat-conduction formulas and boundary conditions [17,18].
(1)$$\rho_{i}\cdot c_{i}\cdot \frac{\partial T\left(z,t\right)}{\partial t}=k_{i}\cdot \frac{\partial^{2}T\left(z,t\right)}{\partial z^{2}},\;0\leq t\leq \tau$$
(2)$$-k_{1}\cdot \frac{\partial T_{1}\left(z,t\right)}{\partial z}|z=0=I_{1}=A_{1}\cdot I_{0}$$
(3)$$\frac{\partial T_{s}\left(z,t\right)}{\partial z}|z=-L_{1}-L_{2}-L_{s}=0$$
(4)$$T_{1}\left(-L_{1},t\right)=T_{2}\left(-L_{1},t\right)$$
(5)$$T_{2}\left(-L_{1}-L_{2},t\right)=T_{s}\left(-L_{1}-L_{2},t\right)$$
(6)$$T\left(z,0\right)=T_{0}$$
where ρ is the density, c is the heat capacity, k is the thermal conductivity, A is the laser absorptivity, I is the peak power density, and T0 is the environment temperature. A laser is applied to the material surface in heat flux, and its power distribution function is shown in Formula (7). The moving periodic pulse function comprises a Gaussian light source and regular operation. The repetition frequency was set to 10 kHz., and the laser pulse duration was 100 ns.
(7)$$I=\{\begin{array}{cc} A\cdot P\cdot {\left(f\cdot \tau \cdot {\pi r}^{2}\right)}^{-1}\cdot \exp\left(-2\cdot \frac{{\left(x-v_{x}t\right)}^{2}-y^{2}}{r^{2}}\right) & \left(N-1\right)\cdot t_{p}<t<\left(N-1\right)\cdot t_{p}+\tau \;\\ 0 & \left(N-1\right)\cdot t_{p}+\tau <t<N\cdot t_{p} \end{array}$$

### 2.3. Design of Experiment and Analytical Methods

The laser’s peak power density and scanning speed are the main factors that influence the temperature-field distribution of the material [19] and further influence the ablation depth of the surface oxide. This section designed a central composite test to optimize these two parameters with Comsol Multiphysics software. The experiment selected four cubic points, five center points of the cube, and four axis points and used Minitab software to generate 13 groups of tests automatically. Based on the peak power density and scanning speed commonly used in actual work, the experiment factor settings are shown in Table 2.

To establish a quadratic single-removal volume-prediction model, Minitab software 6.0 was used to conduct a regression analysis, variance analysis, response surface, and isoline-map analysis on the test results. A white-light interferometer was used to measure the cleaning profile to verify the accuracy of the removal-volume model. In this study, the removal-volume prediction model of Fe_2_O_3_ (H1) and Fe_3_O_4_ (H2) was established to analyze the stepwise removal process based on layered corrosion oxides.

## 3. Experimental Results and Discussion

### 3.1. Distribution of Material Temperature Field

The maximum temperature of each material layer under different laser parameters in the simulation process is shown in Figure 4 and Figure 5.

As seen from Figure 4, when the overlapping rate between adjacent spots was 50%, the maximum temperature of each layer of material gradually increased with the laser peak power density. For Fe_2_O_3_, when the peak power density was 2.5 × 10^6^ W/cm^2^, the maximum surface temperature reached 3014.32 °C, which exceeded the evaporation temperature, and the material phase changed. The peak power density increased to 5 × 10^7^ W/cm^2^, and the maximum temperature of the surface reached 59,776.16 °C. The reason is that the thermal conductivity of the Fe_2_O_3_ layer is extremely low, laser-pulse energy acts at a nanosecond level, and the power accumulates on the surface in a short time, not transmitting and diffusing in time. Hence, the temperature rose to near 60,000 °C instantaneously.

The rust layer begins to melt and decompose when the maximum surface temperature exceeds its melting or evaporation temperature. For example, Fe_2_O_3_ will spoil and generate O_2_ and Fe_3_O_4_. While gasifying, the decomposition product O_2_ will also take away some oxides [20]. Therefore, when the maximum surface temperature of the material exceeds its melting or evaporation temperature, the subsequent temperature rise is of no real significance.

The size of the scanning speed is mainly reflected in the overlapping rate between adjacent spots. As seen in Figure 5, when the peak power density was fixed at 2.5 × 10^6^ W/cm^2^, the maximum temperature on the surface of each material decreased gradually. For Fe_2_O_3_, when the scanning speed was 500 mm/s, its maximum temperature was 3150.97 °C and the top temperature dropped to 2764.09 °C when the scanning speed increased to 2000 mm/s. This is because the overlapping rate between adjacent spots decreased with the addition of scanning speed.

In the cleaning process, the maximum temperature was small because the Fe_3_O_4_ and matrix were far away from the laser-acting area. It did not exceed its phase transition temperature, so the temperature rise had little effect.

### 3.2. Removal Test Results and Prediction Model

According to the simulation results, the ablation depth under each laser-cleaning parameter was obtained, as shown in Table 3. The corresponding simulation results of each parameter are shown in Figure 6.

#### 3.2.1. Regression Analysis and Variance Analysis

Based on the center composite test design, the effects of laser peak power density and scanning speed on ablation depth were investigated and the corresponding quadratic regression model was built. The regression coefficients estimated by coding units in the quadratic regression model are shown in Table 4.

Table 4 shows that the *p*-values were all less than 0.05, corresponding to the primary and secondary main effects, so their influence was significant.

According to the variance analysis in Table 5, the two determination coefficients, R-sq and R-sq (Adjustment), were 98.56% and 97.53%, respectively. The gap between them was small, close to 1, showing a high regression, indicating that the established ablation depth model was good and the regression equation was not misfitted.

The “*” of the F value indicates that the F value reaches 0.05 level significant, that is, the F value reaches significant. *p* value is an indicator to measure the difference between the control group and the experimental group. “*” means that *p* value is less than 0.05, indicating that there is a significant difference between the two groups.

#### 3.2.2. Prediction Model of Removal Volume

The quadratic-regression equation of Fe_2_O_3_-layer ablation depth can be obtained through regression analysis and variance analysis, namely, the single-removal quantity-prediction model.
(8)$$H_{1}=-26.8+4.2*10^{6}*I+1.259*10^{2}*v+5.55*10^{12}{*I}^{2}+6*10^{6}{*v}^{2}-1.350*10^{8}*I*v+\zeta$$
where H1 is the ablation depth of the Fe_2_O_3_ layer, I is the laser peak power density, v is the scanning speed, and ζ is the error. As for the Fe_3_O_4_ layer, its properties differ from those of the Fe_2_O_3_ layer. In addition, based on the central composite test design, the prediction model of the single removal quantity obtained is shown as Formula 2.
(9)$$H_{2}=-13.2+1.29*10^{6}*I+7.44*10^{3}*v+3.0*10^{12}{*I}^{2}+3*10^{6}{*v}^{2}-7.1*10^{9}*I*v+\zeta$$
where H2 is the ablation depth of the Fe_3_O_4_ layer, I is the laser peak power density, v is the scanning speed, and ζ is the error.

### 3.3. Analysis of Stepwise Removal Process for Layered Rust Oxides

According to the composition of the oxide layer, the actual laser-cleaning process can be divided into three stages: (I) Cleaning the Fe_2_O_3_ layer, (II) cleaning the Fe_2_O_3_ and Fe_3_O_4_ layers simultaneously, and (III) cleaning Fe_3_O_4_ layer. The single removal volume of stage 1 and stage 3 follow Formulas 8 and 9, respectively. When laser cleaning is used to clean two layers of materials simultaneously, it is assumed that the total absorbed energy is E, and the energy absorbed by Fe_2_O_3_ is E1. The energy absorbed by Fe_3_O_4_ is E2, then the single-removal volume-prediction model is H3, as shown in Formula 10.
(10)$$H_{3}={\alpha *H}_{1}+{\beta *H}_{2}=\frac{E_{1}}{E}{*H}_{1}+\frac{E_{2}}{E}{*H}_{2}=\frac{E_{1}}{E_{1}+E_{2}}{*H}_{1}+\frac{E_{2}}{E_{1}+E_{2}}{*H}_{2}$$

Figure 7 shows the flow chart of laser cleaning for rust-layered oxides. Imax, v*, L_1_, and L_2_ input values represent the initial peak power density, initial scanning speed, and thickness of the Fe_2_O_3_ layer and Fe_3_O_4_ layer, respectively. Hi represents the total removal volume of laser cleaning. Δh represents the single removal volume of each stage. I* represents the peak power density adjusted during the last cleaning. The initial values of I, h_0_, and Δh were 0.

Based on the thickness L_1_ and L_2_ of the Fe_2_O_3_ and Fe_3_O_4_ layers, the initial peak power density Imax and scanning speed v* were input. Firstly, the relationship between the total removal volume and the total thickness of the oxide layers was determined. When the total removal volume was less than the thickness of the oxide layer, the cleaning process was selected according to other judgment conditions. Case 1: When the residual thickness of Fe_2_O_3_ was more significant than its single removal volume H1, the removal process was stage (I) and Formula (8) was calculated. Case 2: When the residual thickness of Fe_2_O_3_ was less than its single removal volume H1, the removal process was stage II and Formula (10) was calculated. Case 3: When the residual thickness of Fe_3_O_4_ was more significant than its single removal volume H2 after the Fe_2_O_3_ layer was obliterated, the removal process was stage III and Formula (9) was calculated. Case 4: When the residual thickness of Fe_3_O_4_ was less than its single removal volume H2, the initial peak power density needed to be adjusted from Imax to I* according to Formula (9) so that the single removal amount under the initial peak power density and scanning speed was H3*, that is, to ensure the oxide layer on the matrix surface was entirely removed the last time. The corresponding peak power density I, scanning speed v, cleaning depth hi, and cleaning times i were output in each run of Case 1 to Case 4. The operation was stopped when the oxide layer was removed, and final cleaning parameters and cleaning times were output.

## 4. Verification

To verify the accuracy of the prediction model, three different parameters were selected to conduct a single cleaning experiment for each layer (Fe_2_O_3_, Fe_3_O_4_) of materials. Carbon steel with a size of 20 mm × 20 mm was selected as the sample. The sample morphology after cleaning is shown in Figure 8. The white-light interferometer was used to measure the contour of the cleaning groove, and the polynomial was used for fitting, as shown in Figure 9.

The actual measured and theoretically calculated values were compared and analyzed, as shown in Table 6. As can be seen from Table 6, there was a specific difference between the measured value and the theoretical value. However, with the increased ablation depth, the measured value was closer to the theoretical value. All the errors between the test results and the theoretical calculation values were within a reasonable range, proving that the established single-removal volume-prediction model of rusty oxide was effective.

Based on the established prediction model of single removal volume and the cleaning flow chart of the layered rust model, the actual mining samples were cleaned by laser for verification. The morphology of the selected sample and the thickness of the dirt layer are shown in Figure 10. Firstly, parameters with a peak power density of 3.5 × 10^6^ W/cm^2^ and scanning speed of 1000 mm/s were used for cleaning twice. After cleaning, the surface morphology and remaining thickness of dirt were determined, as shown in Figure 11. The residual dirt was only Fe_3_O_4_, and the thickness of the dirt layer was only 20 μm. The laser parameters were adjusted to the peak power density of 3.7 × 10^6^ W/cm^2^ and the scanning speed to 1500 mm/s according to the single-removal volume-prediction model. In this process, the single removal volume was 10.5 μm, so the remaining oxide layer could be removed entirely after cleaning twice.

Based on adjusted laser parameters, the surface morphology of the Fe_3_O_4_ layer after cleaning twice was determined, as shown in Figure 12. It can be seen that the sample surface began to expose the metal matrix, and only a small amount of particulate matter remained on the metal surface, which showed a good cleaning effect. According to the above experimental results, it was proven that the flow chart of laser cleaning layered rust oxides can be used to guide actual cleaning work.

## 5. Conclusions

In this paper, based on the properties of the rusty oxide, the prediction model for a single removal volume of oxides was established, which can guide the efficient and precise cleaning of the oxide without damaging the surface of the substrate. The main conclusions can be listed as follows:
(1)The main dirt on the surface of engineering machinery was layered oxide, the composition of the surface layer was Fe_2_O_3_, and the design of the bottom surface layer was Fe_3_O_4_.(2)A central composite test of peak power density and scanning speed was designed with the help of Comsol Multiphysics software. In this process, the prediction model test of single removal volume for different oxides was established, and experiments proved the model.(3)Based on the response surface and contour map, when the laser energy exceeded the material cleaning threshold, the ablation depth of oxide increased with the peak power density and the decrease in scanning speed.(4)Based on the established prediction model of single removal volume, the stepwise removal process of rust-layered oxides was analyzed. The oxide composition to be cleaned was divided into three stages: (I) cleaning the Fe_2_O_3_ layer, (II) cleaning the Fe_2_O_3_ and Fe_3_O_4_ layers simultaneously, and (III) cleaning Fe_3_O_4_ layer.


## Figures and Tables

**Figure 1 materials-15-07559-f001:**
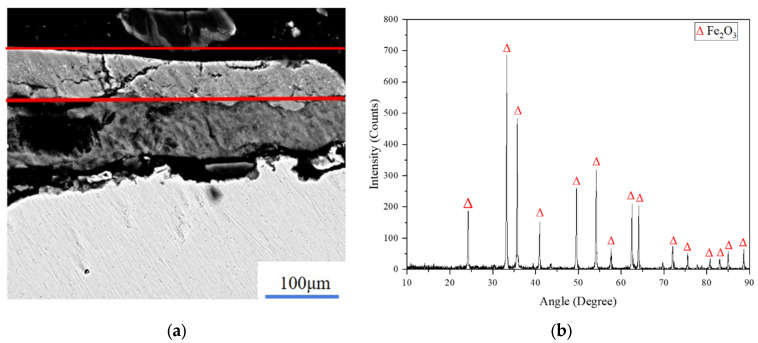
The layered structure distribution of Fe_2_O_3_ and X-ray-diffraction analysis. (**a**) Distribution of Fe_2_O_3_; (**b**) X-ray-diffraction analysis of Fe_2_O_3_.

**Figure 2 materials-15-07559-f002:**
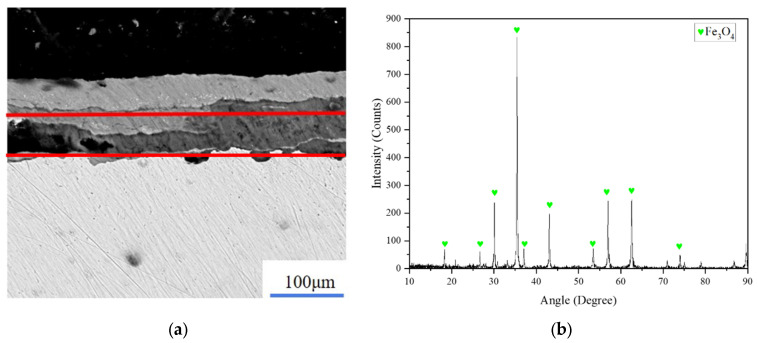
The layered structure distribution of Fe_3_O_4_ and X-ray-diffraction analysis. (**a**) Distribution of Fe_3_O_4_; (**b**) X-ray-diffraction analysis of Fe_3_O_4_.

**Figure 3 materials-15-07559-f003:**
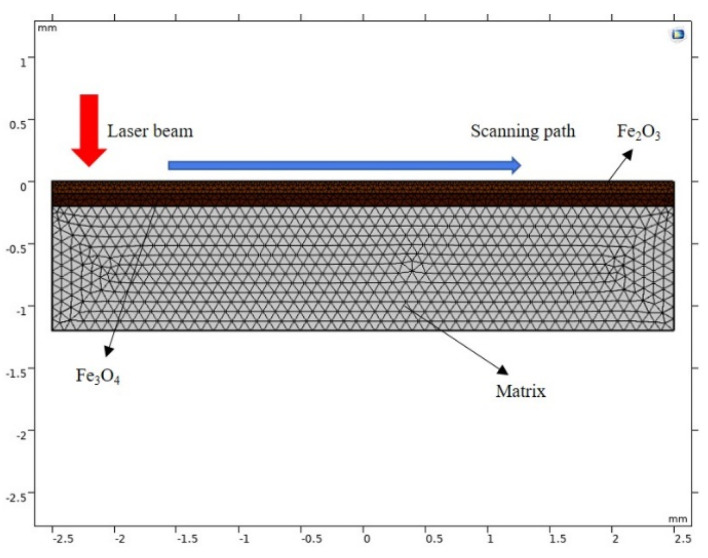
Layered model of cleaning oxide.

**Figure 4 materials-15-07559-f004:**
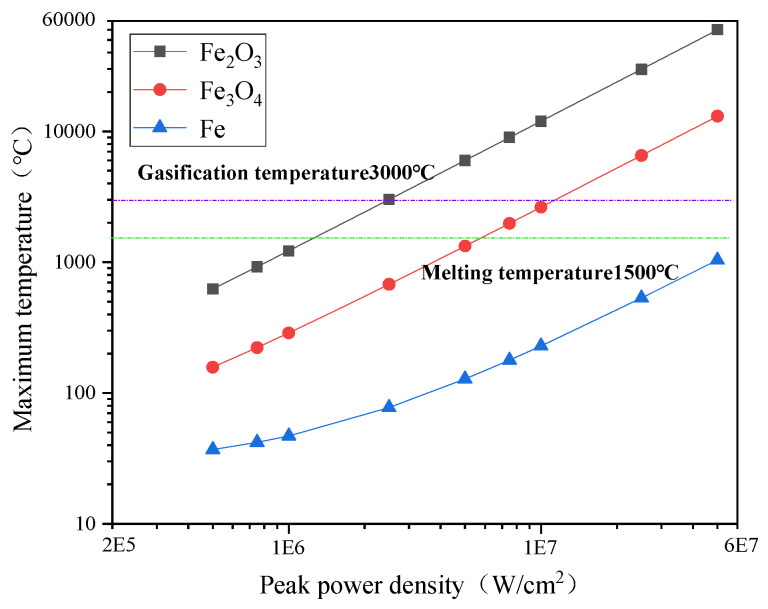
The relationship between the temperature of each layer material and the peak power.

**Figure 5 materials-15-07559-f005:**
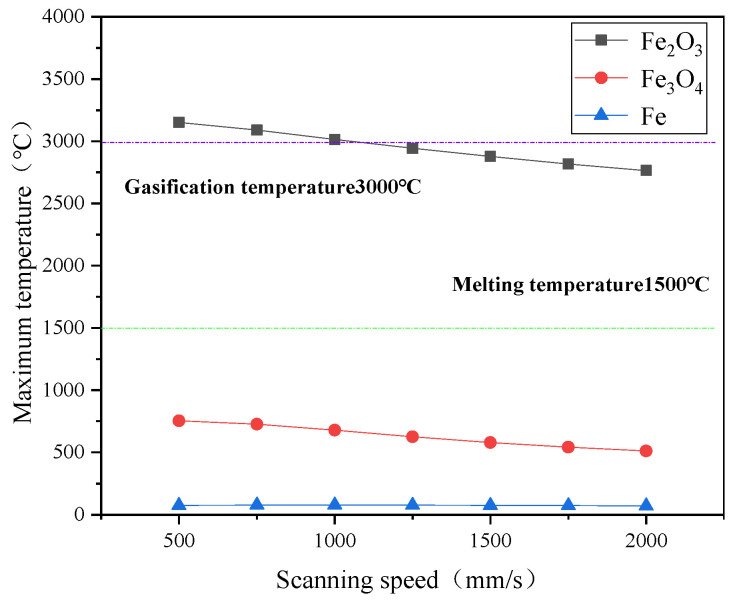
The relationship between the temperature of each layer of material and the scanning speed.

**Figure 6 materials-15-07559-f006:**
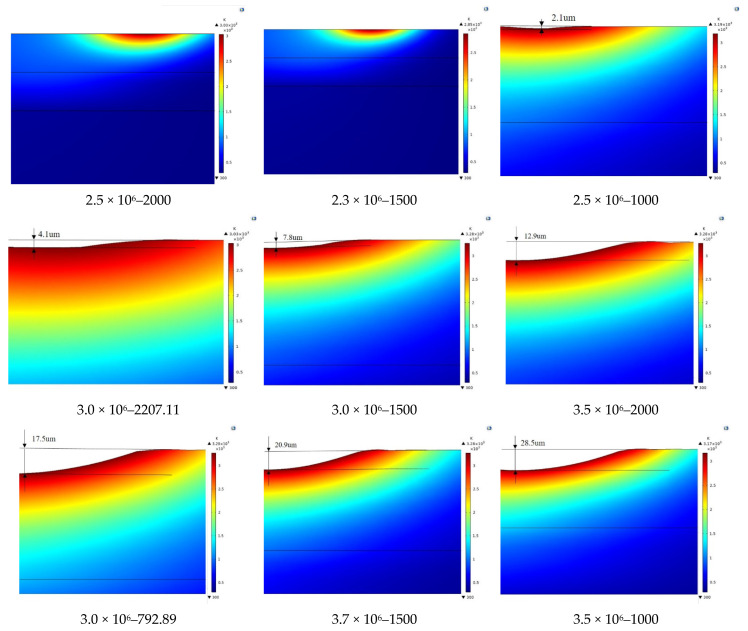
Ablation depth under different laser parameters.

**Figure 7 materials-15-07559-f007:**
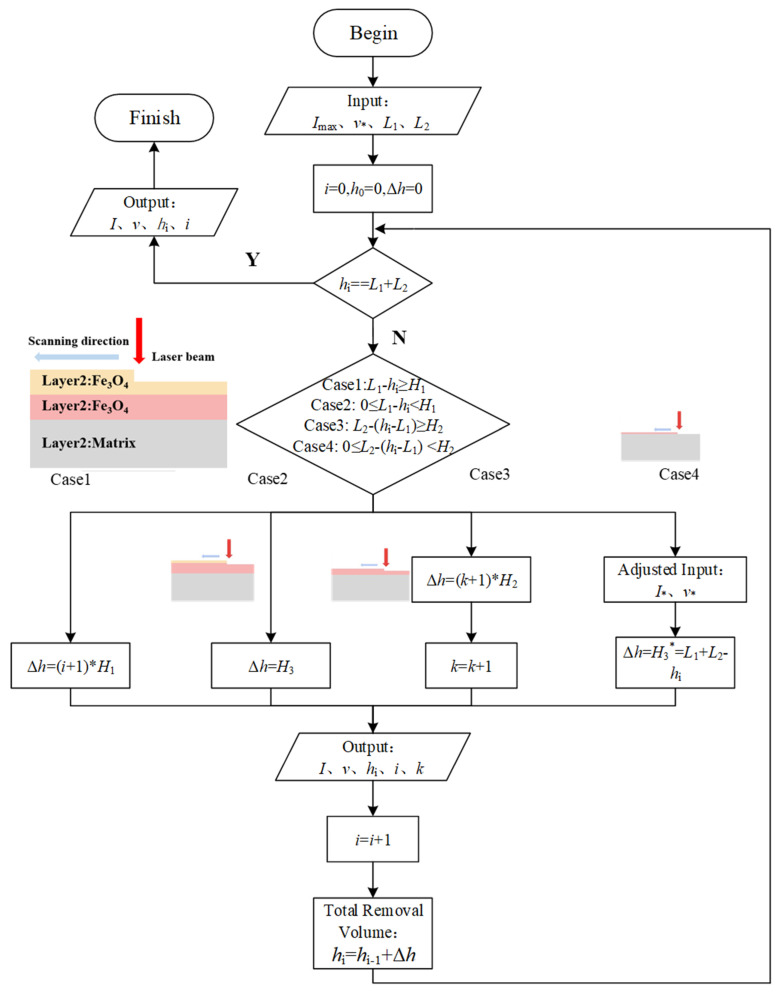
Flow chart of laser cleaning for layered rust oxides.

**Figure 8 materials-15-07559-f008:**
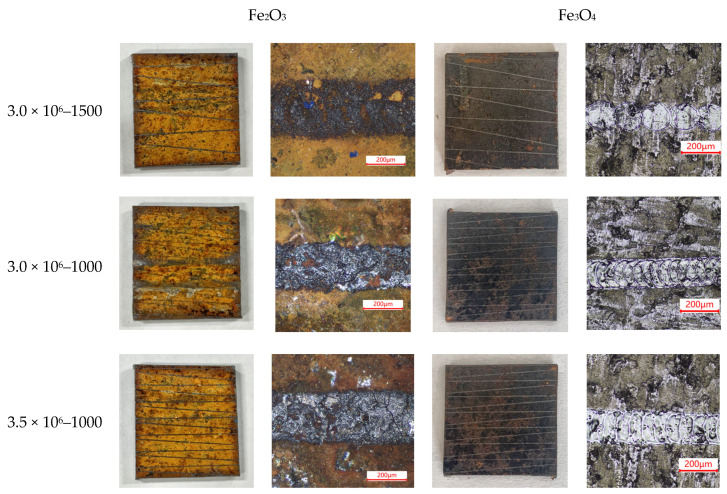
The surface morphology of each layer was cleaned by a single laser time.

**Figure 9 materials-15-07559-f009:**
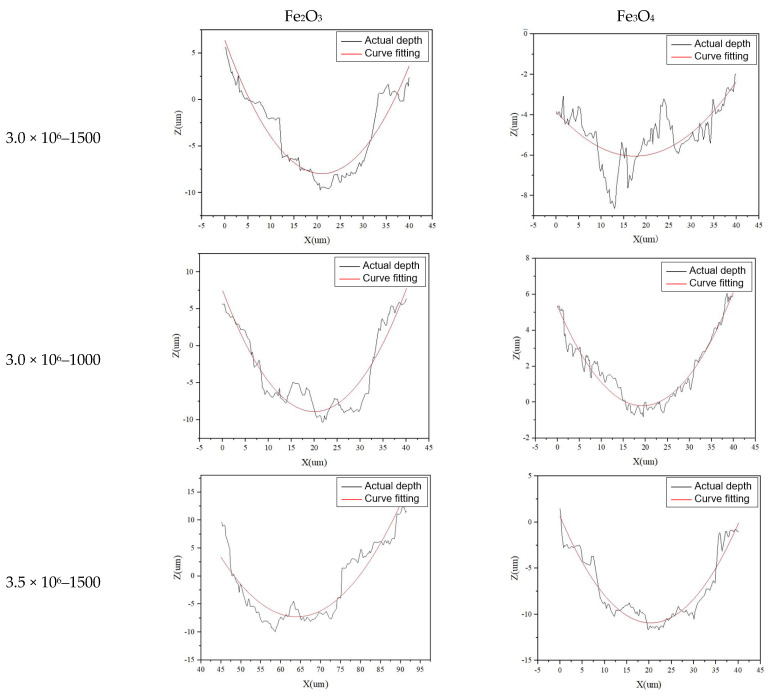
The groove contour after cleaning under different parameters.

**Figure 10 materials-15-07559-f010:**
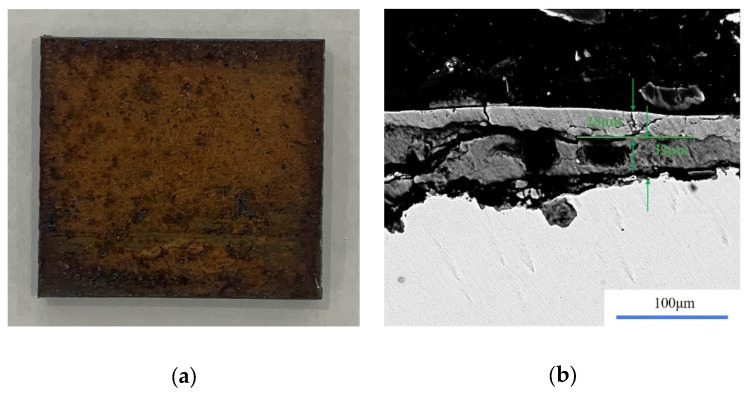
Morphology of sample to be cleaned and thickness of oxide layer. (**a**) Morphology of sample to be cleaned; (**b**) thickness of oxide layer.

**Figure 11 materials-15-07559-f011:**
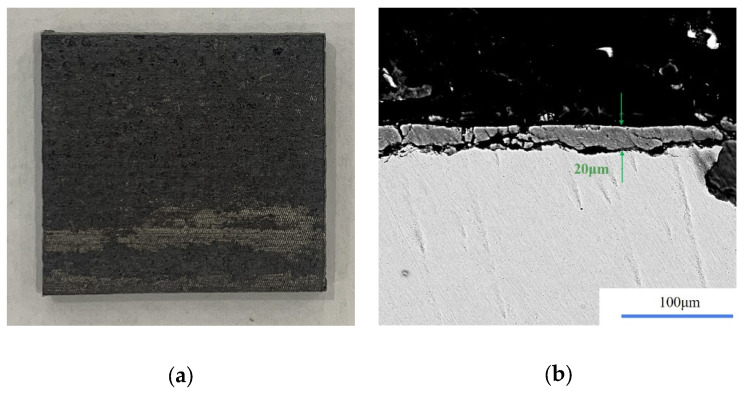
Morphology and thickness of residual dirt after cleaning twice. (**a**) Morphology of sample after cleaning; (**b**) thickness of residual oxide.

**Figure 12 materials-15-07559-f012:**
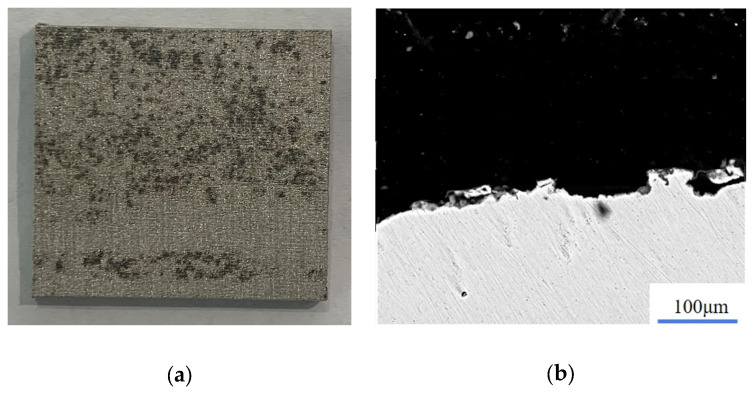
Surface morphology after complete cleaning. (**a**) Morphology of sample after cleaning twice; (**b**) thickness of residual oxide.

**Table 1 materials-15-07559-t001:** Material properties of each oxide.

Material Properties	Fe (Q345)	Fe_2_O_3_	Fe_3_O_4_
Density/ρ (g/cm^3^)	7.86	5.24	5.18
Specific heat capacity/c (J/(Kg·K))	920	626	1350
Thermal conductivity/K (W/(cm·K))	0.52	0.04	0.02
Melting temperature/Tr (°C)	1535	1565	1597
Evaporation temperature /Tv (°C)	2750	2700	3000
Laser absorptivity/A	0.35	0.60	0.53

**Table 2 materials-15-07559-t002:** Horizontal settings of experiment factors.

The Serial Number	Name of Factor	Low Level	High Level
1	Peak power density (W/cm^2^)	2.5 × 10^6^	3.5 × 10^6^
2	Scanning speed (mm/s)	1000	2000

**Table 3 materials-15-07559-t003:** Ablation depth of the layered model under various laser parameters.

Serial Number	Peak Power Density (W/cm^2^)	Scanning Speed (mm/s)	Ablation Depth (μm)
1	2.5 × 10^6^	2000.00	0.0
2	2.3 × 10^6^	1500.00	0.0
3	2.5 × 10^6^	1000.00	2.1
4	3.0 × 10^6^	2207.11	4.1
5	3.0 × 10^6^	1500.00	7.8
6	3.0 × 10^6^	792.89	17.5
7	3.5 × 10^6^	2000.00	12.9
8	3.7 × 10^6^	1500.00	20.9
9	3.5 × 10^6^	1000.00	28.5

**Table 4 materials-15-07559-t004:** Estimated regression coefficient of ablation depth.

Item	Coefficient	Coefficient Standard Error	T	P
Constant	7.800	0.590	13.22	0.000
Peak power density I	8.607	0.466	18.45	0.000
Scanning speed v	−4.581	0.466	−9.82	0.000
Peak power density I* Peak power density I	1.388	0.500	2.77	0.028
Scanning speed v* Scanning speed v	1.563	0.500	3.12	0.017
Peak power density I* Scanning speed v	−3.375	0.660	−5.12	0.001

**Table 5 materials-15-07559-t005:** Variance analysis of ablation depth. R-sq = 98.56% R-sq (adjustment) = 97.53% R-sq (prediction) = 89.75%.

Source	Degrees of Freedom	Adj SS	Adj MS	F	P
Model	5	833.031	166.606	95.70	0.000
Linear	2	760.569	380.284	218.45	0.000
Peak power density I	1	592.662	592.662	340.44	0.000
Scanning speed v	1	167.907	167.907	96.45	0.000
Square	2	26.899	13.450	7.73	0.017
Peak power density I* Peak power density I	1	13.392	13.392	7.69	0.028
Scanning speed v* Scanning speed v	1	16.984	16.984	9.76	0.017
Two-factor interaction	1	45.562	45.562	26.17	0.001
Peak power density I* Scanning speed v	1	45.562	45.562	26.17	0.001
Error	7	12.186	1.741		
Loss of quasi	3	12.186	4.062	*	*
Pure error	4	0.000	0.000		
Total	12	845.217			

“*” in Source stands for multiplication.

**Table 6 materials-15-07559-t006:** Comparative analysis of measurements and under different parameters.

Parameters	3.0 × 10^6^–1500		3.0 × 10^6^–1000		3.5 × 10^6^–1500	
Layers	Fe_2_O_3_	Fe_3_O_4_	Fe_2_O_3_	Fe_3_O_4_	Fe_2_O_3_	Fe_3_O_4_
M-value	8.400	2.900	14.200	6.300	17.500	9.400
T-value	7.385	3.630	13.840	6.810	17.398	8.700
Error	13.744%	20.110%	2.601%	7.489%	0.586%	8.046%

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
