# Peer review of "Stepwise Removal Process Analysis Based on Layered Corrosion Oxides"

_materials, 2022, doi:10.3390/ma15217559_

Round 1

Reviewer 1 Report

The manuscript by Ren et al. reported the cleaning of steel surface from rusty oxides by analyzing laser processing parameters (peak intensity and scanning speed) to avoid damaging the surface due to controlled ablation depth of Fe2O3 and Fe3O4. These results are interest to readers as a technical reference. However, several problems were found and must be clarified to improve the paper's quality. 

1)                In figures 1 and 2 there are several non-attributed XRD peaks. Some comments should be given about it.

2)                Authors solve one-dimensional heat conduction task with appropriate boundary conditions. The laser energy has Gaussian distribution. Is it possible to get better results in terms of surface quality after cleaning with the use of flat-top distribution?

3)                Figures 4 and 5 seems to be unnecessary, while figures 3 and 6 could be merged.

4)                For figures 7 and 8 it is better to indicate the scanning speed and peak intensity, correspondingly, at figures’ caption.

5)                It is better to say evaporation temperature instead of gasification temperature.

6)                Figure 9 must be modified to clearly show an ablation depth and temperature color scale.

7)                Figure 10 demonstrates a flow chart of laser cleaning for layered rust oxides. However, its content is duplicated in the text at the end of p. 16.

8)                The scale bars in figure 11 are missed or unclear.

9)                What is the accuracy of measurements presented in figure 12?

10)           Some issues could be found in the text:

a.      Row 160 – “Bulleted lists look like this:”

b.     The spacings between a value and its unit of measure are missed.

c.      Row 241 – “The cleaning process is:”

d.     Careful proofreading is highly recommended.

Author Response

Thank you very much for your comments on our manuscript. We want to express our sincere appreciation for your careful reading and invaluable comments. These comments are precious and helpful in improving our manuscripts, and they have important guiding significance for our thesis writing and research work. We have carefully revised the manuscript based on your comments. Please see the attachment.

Reviewer 2 Report

This will review the “Stepwise removal process analysis based on layered corrosion 2 oxides“. The subject has a lot to do with industry and can be used in industrial environments. The manuscript is well designed and written.

1.     In the introduction section, describe the mechanism of laser cleaning. How does laser remove oxide layers? This will be useful for those who are familiar with this method for the first time.

2.     Explain the difference between pulsed and continuous laser in laser cleaning.

3.     How did you calculate the overlap percentage in line 168?

Author Response

(The authors gave the same response as above.)

Round 2

Reviewer 1 Report

Authors have addressed the most of my comments, the paper can be published now.